# Developing and validating subjective and objective risk-assessment measures for predicting mortality after major surgery: An international prospective cohort study

**Danny J. N. Wong** [1,2], **Steve Harris** [3], **Arun Sahni** [1,2], **James R. Bedford** [1,2], **Laura Cortes** [2], **Richard Shawyer** [4], **Andrew M. Wilson** [5], **Helen A. Lindsay** [5], **Doug Campbell** [5], **Scott Popham** [6], **Lisa M. Barneto** [7], **Paul S. Myles** [8], **SNAP-2: EPICCS collaborators** [¶], **S. Ramani Moonesinghe** [1,2] *

1 UCL/UCLH Surgical Outcomes Research Centre, Centre for Perioperative Medicine, Department for Targeted Intervention, Division of Surgery and Interventional Science, University College London, London, United Kingdom, 2 Health Services Research Centre, National Institute of Academic Anaesthesia, Royal College of Anaesthetists, London, United Kingdom, 3 Bloomsbury Institute of Intensive Care Medicine, University College London, London, United Kingdom, 4 Lay representative, United Kingdom, 5 Auckland City Hospital, Auckland District Health Board, Auckland, New Zealand, 6 Gold Coast University Hospital, Southport, Queensland, Australia, 7 Wellington Regional Hospital, Capital & Coast District Health Board, Wellington, New Zealand, 8 Department of Anaesthesiology and Perioperative Medicine, The Alfred Hospital, Melbourne, Victoria, Australia

¶ A full list of collaborators and their affiliations is included in the supporting information (S12 Text).
* ramani.moonesinghe@nhs.net

**Data Availability Statement:** All relevant data are within the manuscript and its supporting information files.

## Abstract

### Background

Preoperative risk prediction is important for guiding clinical decision-making and resource allocation. Clinicians frequently rely solely on their own clinical judgement for risk prediction rather than objective measures. We aimed to compare the accuracy of freely available objective surgical risk tools with subjective clinical assessment in predicting 30-day mortality.

### Methods and findings

We conducted a prospective observational study in 274 hospitals in the United Kingdom (UK), Australia, and New Zealand. For 1 week in 2017, prospective risk, surgical, and outcome data were collected on all adults aged 18 years and over undergoing surgery requiring at least a 1-night stay in hospital. Recruitment bias was avoided through an ethical waiver to patient consent; a mixture of rural, urban, district, and university hospitals participated. We compared subjective assessment with 3 previously published, open-access objective risk tools for predicting 30-day mortality: the Portsmouth-Physiology and Operative Severity Score for the enUmeration of Mortality (P-POSSUM), Surgical Risk Scale (SRS), and Surgical Outcome Risk Tool (SORT). We then developed a logistic regression model combining subjective assessment and the best objective tool and compared its performance to each constituent method alone. We included 22,631 patients in the study: 52.8% were female,

**Funding:** National Institute for Academic Anaesthesia (NIAA) Association of Anaesthetists project grant awarded to SRM: www.niaa.org.uk. The funders had no role in study design, data collection and analysis, decision to publish, or preparation of the manuscript.

**Competing interests:** The authors have declared that no competing interests exist.

**Abbreviations:** ASA-PS, American Society of Anesthesiologists Physical Status; AUROC, Area Under Receiver Operating Characteristic curve; CARES, Combined Assessment of Risk Encountered in Surgery; CI, confidence interval; COPD, Chronic Obstructive Pulmonary Disease; CPET, cardiopulmonary exercise testing; DCA, decision-curve analysis; IQR, interquartile range; METS, Metabolic Equivalents; NRI, Net Reclassification Improvement; P-POSSUM, Portsmouth-Physiology and Operative Severity Score for the enUmeration of Mortality; QI, Quality Improvement; SNAP-2: EPICCS, Second Sprint National Anaesthesia Project: EPIdemiology of Critical Care provision after Surgery; SORT, Surgical Outcome Risk Tool; SRS, Surgical Risk Scale; STROBE, Strengthening the Reporting of Observational Studies in Epidemiology; TRIPOD, Transparent Reporting of a multivariable prediction model for Individual Prognosis Or Diagnosis; VISION, Vascular Events in Non-cardiac Surgery patients cohort study.

median age was 62 years (interquartile range [IQR] 46 to 73 years), median postoperative length of stay was 3 days (IQR 1 to 6), and inpatient 30-day mortality was 1.4%. Clinicians used subjective assessment alone in 88.7% of cases. All methods overpredicted risk, but visual inspection of plots showed the SORT to have the best calibration. The SORT demonstrated the best discrimination of the objective tools (SORT Area Under Receiver Operating Characteristic curve [AUROC] = 0.90, 95% confidence interval [CI]: 0.88–0.92; P-POSSUM = 0.89, 95% CI 0.88–0.91; SRS = 0.85, 95% CI 0.82–0.87). Subjective assessment demonstrated good discrimination (AUROC = 0.89, 95% CI: 0.86–0.91) that was not different from the SORT (p = 0.309). Combining subjective assessment and the SORT improved discrimination (bootstrap optimism-corrected AUROC = 0.92, 95% CI: 0.90–0.94) and demonstrated continuous Net Reclassification Improvement (NRI = 0.13, 95% CI: 0.06–0.20, p < 0.001) compared with subjective assessment alone. Decision-curve analysis (DCA) confirmed the superiority of the SORT over other previously published models, and the SORT–clinical judgement model again performed best overall. Our study is limited by the low mortality rate, by the lack of blinding in the 'subjective' risk assessments, and because we only compared the performance of clinical risk scores as opposed to other prediction tools such as exercise testing or frailty assessment.

## Conclusions

In this study, we observed that the combination of subjective assessment with a parsimonious risk model improved perioperative risk estimation. This may be of value in helping clinicians allocate finite resources such as critical care and to support patient involvement in clinical decision-making.

## Author summary

### Why was this study done?

- Over 3 million postoperative deaths occur worldwide per year.
- Some of these may be avoidable through risk-assessment–based modification of treatment pathways, such as postoperative critical care admission.
- There are multiple methods for predicting which patients are at high risk of death or complications from surgery, but these are not widely used, with clinicians instead usually relying on their subjective clinical judgement alone.
- Before this study, there was little information about whether clinical judgement was of better, worse, or equivalent accuracy to objective risk scores.

### What did the researchers do and find?

- We conducted a 1-week cohort study in 274 hospitals in the UK, Australia, and New Zealand, during which we collected data on risk and surgical outcome on every patient who had an operation requiring an overnight stay in hospital.

- The clinical team (surgeons, anaesthetists) looking after the patient were asked to provide a subjective assessment of risk. We compared these assessments with the results of 3 freely available objective risk-assessment tools.

- We included data from 22,631 patients in our analyses and found that subjective assessment was as accurate as the best of the objective risk tools (the Surgical Outcome Risk Tool or SORT) for predicting death in hospital within 30 days of surgery.

- However, combining subjective and objective measurement using the SORT provided an even more accurate estimate.

### What do these findings mean?

- The new SORT–clinical judgement calculator can be used by clinicians to risk-stratify patients and so identify which individuals are most likely to benefit from limited resources such as access to postoperative critical care.

- At the policy level, the tool can be used to plan surgical services, including the number of critical care beds required to serve a surgical population.

- This study is limited by being conducted solely in high-income countries, limiting its global generalisability.

## Introduction

The provision of safe surgery is an international healthcare priority [1]. Guidelines recommend that preoperative risk estimation should guide treatment decisions and facilitate shared decision-making [2,3]. Furthermore, there is an ethical imperative (and in the United Kingdom [UK], a legal requirement) to provide an individualised assessment of a patient's risk of adverse outcomes [4]. Increasing evidence suggests that postoperative mortality in both high and low/middle-income settings is due less to what happens in the operating theatre and more to our 'failure to rescue' patients who develop postoperative complications [5,6]. These observations also point towards opportunity: once a patient has been identified as high risk, mitigation strategies such as pre-emptive admission to critical care or enhanced postoperative surveillance may prevent adverse outcomes [2]. However, critical care is a finite resource, with competition for beds between surgical and emergency medical admissions. To that end, the requirement for a postoperative critical care bed is itself a risk factor for last-minute cancellation, with consequent potential for disruption and harm for both patients and healthcare providers [7]. Thus, there is a need to accurately stratify patient risk so as to make the most of limited resources and improve perioperative outcomes. This is especially true given the scale of demand; more than 300 million operations take place annually worldwide [8]. With a major postoperative morbidity rate of around 15% [9,10], a short-term mortality rate between 1 and 3% [11], and a reproducible association between short-term morbidity and long-term survival [9,12,13], the impact of surgical complications on individual patients, healthcare resources, and society at large is clearly evident. Furthermore, if resources permitted, substantially larger numbers of patients would be considered for surgical intervention [1].

There are numerous methods available to help clinicians estimate perioperative risk, including frailty indices [14], functional capacity assessments such as cardiopulmonary

exercise testing (CPET) [15], and dozens of risk prediction scores and models, many of which are open-source, are easily applied, and have been validated in multiple heterogeneous surgical cohorts [16]. Despite this myriad of choices, data from national Quality Improvement (QI) programmes indicate that clinicians do not routinely document an individualised risk assessment before surgery [10,17]. In part, this may relate to the availability of complex investigations and equipoise over which method is most accurate, particularly when the accuracy of objective methods compared with subjective assessment alone is disputed [15]. We therefore performed a prospective cohort study with the following objectives: to describe how clinicians assess risk in routine practice, to externally validate and compare the performance of 3 open-access risk models with subjective assessment, and to investigate whether objective risk tools add value to subjective assessment.

## Methods

This is a planned analysis of the Second Sprint National Anaesthesia Project: EPIdemiology of Critical Care provision after Surgery (SNAP-2: EPICCS) study, a prospective observational cohort study conducted in 274 hospitals from the UK, Australia, and New Zealand [18]. We report our findings in accordance with the Strengthening the Reporting of Observational Studies in Epidemiology (STROBE; S1 Text) and the Transparent Reporting of a multivariable prediction model for Individual Prognosis Or Diagnosis (TRIPOD; S2 Text) statements [19,20]. National research networks, including trainee-led networks, were used to maximise recruitment from public hospitals in all countries. All adult ($\geq$18 years) patients undergoing inpatient surgery and meeting our criteria (see 'Data set', below) during a 1-week period were included in our analyses for this paper. Patients were recruited between 21–27 March 2017 in the UK, 21–27 June 2017 in Australia, and 6–13 September 2017 in New Zealand.

### Ethical and governance approvals

UK-wide ethical approval for the study was obtained from the Health Research Authority (South Central–Berkshire B REC, reference number: 16/SC/0349); additional permission to collect patient-identifiable data without consent was granted through Section 251 exemption from the Confidentiality Advisory Group for England and Wales (CAG reference: 16/CAG/ 0087), the NHS Scotland Public Benefit and Privacy Panel for Health and Social Care (PBPP reference: 1617–0126), and individual Health and Social Care Trust research and development departments for each site in Northern Ireland (Belfast, Northern, South Eastern, and Western Health and Social Care Trusts, IRAS reference number: 154486). In Australia, each state had different regulatory approval processes, and approvals were received from the following ethics committees: New South Wales—Hunter New England and Greater Western Human Research Ethics Committee; Queensland—Metro South Hospital and Health Service Human Research Ethics Committee; South Australia—Southern Adelaide Clinical Human Research Ethics Committee; Tasmania—Tasmania Health and Medical Human Research Ethics Committee; Victoria—Alfred Health, Eastern Health, Goulburn Valley, Mercy Health, Monash Health, Peter MacCallum Cancer Centre Research Ethics Committees; Western Australia—South Metropolitan Health Service Human Research Ethics Committee. In New Zealand, the study received national approval from the Health and Disability Ethics Committees (Ethics ref: 17/ NTB/139).

### Data set

All data (S3 Text) were collected prospectively. In this study, we defined objective risk assessment as the use of a risk calculation model or equation or tool that supplies a prediction of risk

on a probability scale. Before surgery, perioperative teams answered the following question for each patient: 'What is the estimate of the perioperative team of the risk of death within 30 days?', with 6 categorical response options (<1%, 1%–2.5%, 2.6%–5%, 5.1%–10%, 10.1%–50%, and >50%). These thresholds were decided by expert consensus within the study steering group and study authors. Teams were then asked to record how they arrived at this estimate (for example, clinical judgement and/or an objective risk tool). The patient data for this study were collected from a wide range of participating publicly funded hospitals in the UK (n = 245), Australia (n = 21), and New Zealand (n = 8). These were a heterogeneous mix of secondary (42%) and tertiary care (58%) institutions and likely reflective of the general composition of hospitals in these countries. We have previously described the hospitals and their available facilities for providing perioperative care [21].

Patients included in the study were adults (≥18 years) undergoing surgery or other interventions that required the presence of an anaesthetist and who were expected to require overnight stay in hospital. We included all procedures taking place in an operating theatre, radiology suite, endoscopy suite, or catheter laboratory for which inpatient (overnight) stay was planned, including both planned and emergency/urgent surgery of all types, endoscopy, and interventional radiology procedures.

Patients were excluded if they indicated they did not want to participate in the study. We also excluded ambulatory surgery, obstetric procedures (for example, cesarean sections and surgery for complications of childbirth), procedures on ASA-PS (American Society of Anesthesiologists Physical Status score) grade VI patients, noninterventional diagnostic imaging (for example, CT or MRI scanning without interventions), and emergency department or critical care interventions requiring anaesthesia or sedation but no interventional procedure.

## Statistical analysis

The protocol for SNAP-2: EPICCS was previously published with aims, objectives, and research questions outlined [18]. Our primary outcome for the study described in this paper was inpatient 30-day mortality, recorded prospectively by local collaborators. We conducted 3 inferential analyses, the first using the entire patient data set and the second and third omitting the patients for whom an objective tool was used to predict perioperative risk (Fig 1). For the first analysis, we evaluated performance of the Portsmouth-Physiology and Operative Severity Score for the enUmeration of Mortality (P-POSSUM), Surgical Risk Scale (SRS), and Surgical Outcome Risk Tool (SORT) [16,22–24]. The calibration and discrimination of all models was assessed in accordance with the Transparent Reporting of a multivariable prediction model for Individual Prognosis Or Diagnosis (TRIPOD) recommendations [20]. Calibration was assessed by graphical inspection of observed versus expected mortality and by the Hosmer–Lemeshow goodness-of-fit test [25]. Discrimination was assessed by calculating the Area Under Receiver Operating Characteristic curve (AUROC) [26]. AUROCs were compared using DeLong's test for 2 correlated ROC curves [27]. ROC curves can be constructed for both continuous predictions (for example, P-POSSUM, SRS, and SORT) and ordinal categorical predictions (for example, ASA-PS or the 6-category subjective predictions that clinicians were asked to make): in the former, sensitivities and specificities are calculated for every value in the probability range of 0 to 1, and then each point is plotted to obtain a smooth curve; in the latter, sensitivities and specificities are computed for each category, and the points form a polygon on the ROC plot.

The second analysis compared the performance of subjective assessment (defined as either using clinical judgement and/or ASA-PS) against the best-performing risk tool. For this, we included only patients for whom subjective assessment alone was used to predict the risk of

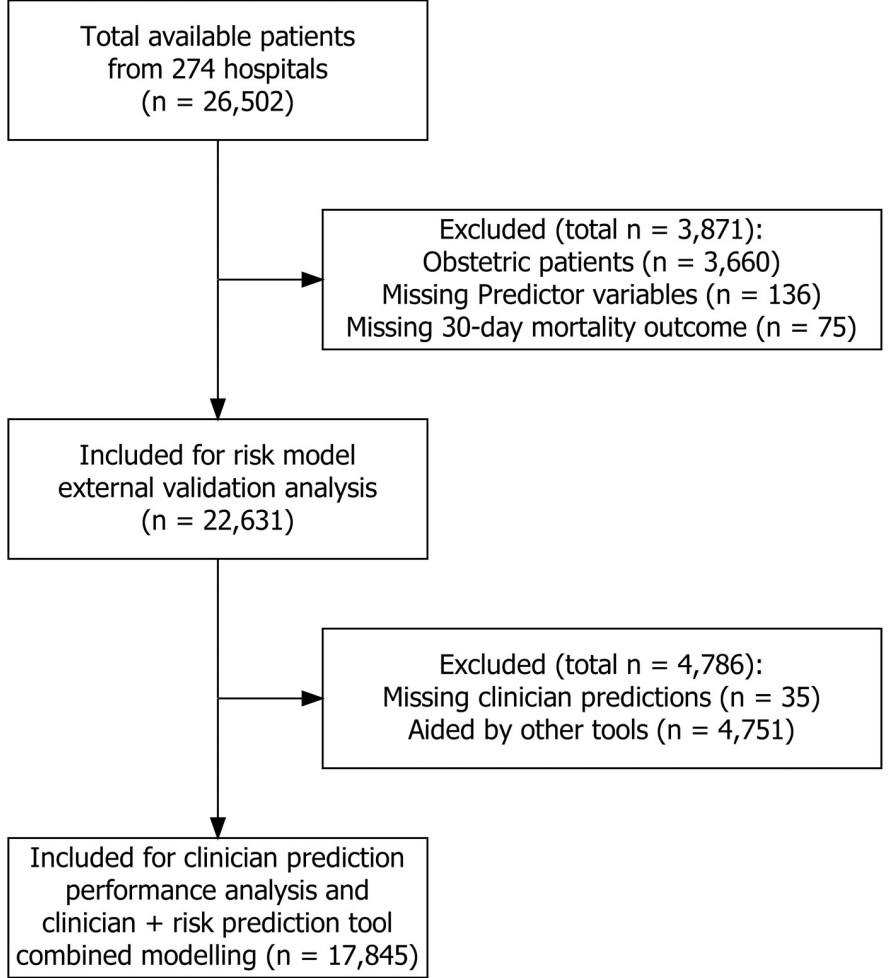

**Fig 1. Participant flowchart.**

30-day mortality. Subjective assessment was then evaluated on calibration and discrimination. Point estimates of risk prediction were taken as the midpoint of the predicted risk intervals provided by clinicians (i.e., 0.5% for the interval <1%, 1.75% for the interval 1%–2.5%, and so on), and the proportion of observed mortality in each of these risk categories was calculated. Calibration was then assessed by plotting the observed mortality proportions against the midpoints of clinician-predicted risk intervals. We then compared the performance of subjective assessment against the best-performing risk model, using AUROC and the continuous Net Reclassification Improvement (NRI) statistic [25]. The NRI quantifies the proportion of individuals whose predictions improve in accuracy (positive reclassification) subtracted by the proportion whose predictions worsened in accuracy (negative reclassification) when using one prediction model versus another [28]. An NRI >0 indicates an overall improvement, <0 an overall deterioration, and zero no difference in prediction accuracy.

The third analysis evaluated the added value of combining subjective assessment with the best-performing risk tool by creating a logistic regression model with variables from both sources.

For this, we fitted a logistic regression model with 2 variables: the subjective assessment of risk and the mortality prediction from the best objective risk tool according to the following

logit formula: $ln(R/(1 − R)) = \beta_0 + \beta_1 X_{subjective} + \beta_2 X_{objective}$, where $R$ is the probability of 30-day mortality; $\beta_0$, $\beta_1$, and $\beta_2$ are the model coefficients; $X_{subjective}$ is the subjective clinical assessment (6 ordered categories, as above); and $X_{objective}$ is the risk of mortality as predicted using the most accurate risk model. An optimism-corrected performance estimate of the combined model was obtained using bootstrapped internal validation with 1,000 repetitions; this was then compared with subjective assessment and the most accurate risk model alone.

We used decision-curve analysis (DCA) to describe and compare the clinical implications of using each risk model. In DCA, a model is considered to have clinical value if it has the highest net benefit across the whole range of thresholds for which a patient would be labelled as 'high risk'. The net benefit is defined as the difference between the proportion of true positives (labelled as high risk and then going on to die within 30 days of surgery) and the proportion of false positives (labelled as high risk but not going on to die within 30 days) weighted by the odds of the selected threshold for the high-risk label. At any given threshold, the model with the higher net benefit is the preferred model [20, 25, 29].

### Missing data

The P-POSSUM requires biochemical and haematological data for calculation; however, fit patients may not have preoperative blood tests [30], and in other cases, there may be no time for blood analysis before surgery. Therefore, in cases for which these data were missing, normal physiological ranges were imputed because this most closely follows what clinicians might reasonably do in practice when tests are not indicated or not feasible or results are missing. Following imputation, we performed a complete case analysis because we considered the proportion of cases with missing data in the remaining variables to be low (1.08%) [31].

### Sensitivity analyses

We conducted a number of sensitivity analyses to examine the potential effects of differences in population characteristics on our main study findings. First, we repeated our analyses in a full cohort of patients, including those undergoing obstetric procedures. Second, we repeated the analysis in a subgroup of high-risk patients, defined according to previously published criteria based on age, type of surgery, and comorbidities [15,32]. Third, we evaluated the impact on the accuracy of subjective assessment of using objective tools by comparing discrimination and calibration of subjective assessment in the subgroup of patients whose risk estimates were not solely informed by clinical judgement. Fourth, we repeated our analyses separately in the UK and Australian/New Zealand cohorts to investigate the potential for geographical influences on our findings. Fifth, we examined the potential impact of normal value imputation on missing P-POSSUM values by repeating the analysis on only cases in which no missing P-POSSUM variables were present. Finally, we conducted analyses on surgical specialty subgroups to evaluate the accuracy of the new model created on different subcohorts.

Analyses were performed using R version 3.5.2; $p < 0.05$ was considered statistically significant. Statistical code is available on request.

## Results

Patient data were collected on 26,502 surgical episodes in 274 hospitals across the UK, Australia, and New Zealand (Table 1). A total of 3,871 cases were excluded from all analyses: 3,660 obstetric cases in which there were no deaths, plus a further 286 cases for missing values. This left 22,631 cases with adequate data for external validation of the P-POSSUM, SRS, and SORT models, the first part of our analyses (Fig 1). For the second and third analyses, in which we compared subjective assessment against the best-performing objective risk tool and combined

**Table 1. Patient demographics stratified by 30-day mortality.**

| | | 30-Day Mortality | |
| --- | --- | --- | --- |
| | *Overall* | *Survived* | *Died* |
| N | 22,631 | 22,314 | 317 |
| Male sex (%) | 10,671 (47.2) | 10,481 (47.0) | 190 (59.9) |
| Female sex (%) | 11,960 (52.8) | 11,833 (53.0) | 127 (40.1) |
| Age (median [IQR]) | 62 [46–73] | 62 [45–73] | 76.00 [64.00–83.00] |
| Operative urgency (%) | | | |
| Elective | 12,061 (53.3) | 12,029 (53.9) | 32 (10.1) |
| Expedited | 3,311 (14.6) | 3,270 (14.7) | 41 (12.9) |
| Urgent | 6,617 (29.2) | 6,460 (29.0) | 157 (49.5) |
| Immediate | 642 (2.8) | 555 (2.5) | 87 (27.4) |
| ASA-PS class (%) | | | |
| I | 4,462 (19.7) | 4,458 (20.0) | 4 (1.3) |
| II | 10,192 (45.0) | 10,168 (45.6) | 24 (7.6) |
| III | 6,574 (29.0) | 6,454 (28.9) | 120 (37.9) |
| IV | 1,337 (5.9) | 1,206 (5.4) | 131 (41.3) |
| V | 66 (0.3) | 28 (0.1) | 38 (12.0) |
| Procedure severity (%)* | | | |
| Minor | 1,951 (8.6) | 1,919 (8.6) | 32 (10.1) |
| Intermediate | 4,523 (20.0) | 4,476 (20.1) | 47 (14.8) |
| Major | 7,478 (33.0) | 7,369 (33.0) | 109 (34.4) |
| Xmajor | 5,281 (23.3) | 5,218 (23.4) | 63 (19.9) |
| Complex | 3,398 (15.0) | 3,332 (14.9) | 66 (20.8) |
| Surgical specialty (%) | | | |
| Gastrointestinal surgery | 4,472 (19.8) | 4,384 (19.6) | 88 (27.8) |
| Gynaecology/urology | 4,309 (19.0) | 4,297 (19.3) | 12 (3.8) |
| Neuro/spinal surgery | 1,208 (5.3) | 1,181 (5.3) | 27 (8.5) |
| Orthopaedics | 6,772 (29.9) | 6,688 (30.0) | 84 (26.5) |
| Thoracic/cardiac surgery | 1,033 (4.6) | 1,015 (4.5) | 18 (5.7) |
| Vascular | 674 (3.0) | 645 (2.9) | 29 (9.1) |
| Other | 4,163 (18.4) | 4,104 (18.4) | 59 (18.6) |
| Past medical history: coronary artery disease (%) | 3,029 (13.4) | 2,923 (13.1) | 106 (33.4) |
| Past medical history: congestive cardiac failure (%) | 893 (3.9) | 839 (3.8) | 54 (17.0) |
| Past medical history: metastatic cancer (active) (%) | 825 (3.6) | 799 (3.6) | 26 (8.2) |
| Past medical history: dementia (%) | 676 (3.0) | 644 (2.9) | 32 (10.1) |
| Past medical history: COPD (%) | 1,955 (8.6) | 1,909 (8.6) | 46 (14.5) |
| Past medical history: pulmonary fibrosis (%) | 180 (0.8) | 173 (0.8) | 7 (2.2) |
| Past medical history: liver cirrhosis (%) | 224 (1.0) | 206 (0.9) | 18 (5.7) |
| Past medical history: renal disease (%) | 381 (1.7) | 362 (1.6) | 19 (6.0) |
| Past medical history: diabetes (%) | | | |
| Type 1 | 274 (1.2) | 265 (1.2) | 9 (2.8) |
| Type 2 (dietary-controlled) | 614 (2.7) | 598 (2.7) | 16 (5.0) |
| Type 2 (insulin-controlled) | 761 (3.4) | 743 (3.3) | 18 (5.7) |
| Type 2 (oral hypoglycaemic medication) | 1,570 (6.9) | 1,522 (6.8) | 48 (15.1) |
| No diabetes | 19,399 (85.8) | 19,173 (86.0) | 226 (71.3) |
| Postoperative length of stay in days (median [IQR]) | 3 [1–6] | 3 [1–6] | 7 [2–13 |
| SORT-calculated mortality risk % (median [IQR]) | 0.4 [0.2–1.6] | 0.4 [0.2–1.6] | 8.9 [4.2–20.6] |
| P-POSSUM-calculated mortality risk % (median [IQR]) | 1.1 [0.6–2.9] | 1.1 [0.6–2.9] | 18.1 [5.7–41.6] |

*(Continued)*

**Table 1.** (Continued)

| | Overall | 30-Day Mortality | |
| --- | --- | --- | --- |
| | | Survived | Died |
| SRS-calculated mortality risk % (median [IQR]) | 1.9 [0.8–4.4] | 1.9 [0.8–4.4] | 19.6 [4.4–36.1] |
| Subjective clinical assessment made on clinical judgement and/or ASA-PS grading alone (%) | 17,845 (78.9) | 17,657 (79.1) | 188 (59.3) |

*Procedure severity classification (minor, intermediate, major, Xmajor, and complex: ordinal scale).

**Abbreviations:** ASA-PS, American Society of Anesthesiologists Physical Status; COPD, Chronic Obstructive Pulmonary Disease; IQR, interquartile range; P-POSSUM, Portsmouth-Physiology and Operative Severity Score for the enUmeration of Mortality; SORT, Surgical Outcome Risk Tool; SRS, Surgical Risk Scale.

these measures to create a new model for internal validation, we excluded 4,891 cases in which clinician prediction was aided by the use of other risk tools. This left 21,325 cases for these analyses. There were 317 inpatient deaths within 30 days of surgery (1.40%). In most cases, subjective assessment alone was used to estimate risk (n = 17,845, 78.9%; Table 2). No patients were lost to follow-up.

## External validation of existing risk prediction models

The SORT was the best calibrated of the pre-existing models; however, all overpredicted risk (Fig 2A–2C; Hosmer–Lemeshow p-values all <0.001 for the SORT, P-POSSUM, and SRS). All models exhibited good-to-excellent discrimination (Fig 2D; AUROC SORT = 0.90, 95% confidence interval [CI]: 0.88–0.92; P-POSSUM = 0.89, 95% CI: 0.88–0.91; SRS = 0.85, 95% CI: 0.82–0.87). The AUROC for the SORT was significantly better than SRS (p < 0.001), but not P-POSSUM (p = 0.298).

**Table 2. Methods used by clinicians to estimate 30-day mortality.** Clinicians could select one or more categories; therefore, the total percentages (in parentheses) exceed 100%.

| | Overall |
| --- | --- |
| n | 22,631 |
| Clinical judgement (%) | 20,064 (88.7) |
| ASA-PS score (%) | 8,622 (38.1) |
| Duke Activity Status Index or other activity index (%) | 515 (2.3) |
| Six-minute walk test or incremental shuttle walk test (%) | 48 (0.2) |
| Cardiopulmonary exercise testing (%) | 215 (1.0) |
| Formal frailty assessment (for example, Edmonton Frail Scale) (%) | 48 (0.2) |
| SRS (%) | 315 (1.4) |
| SORT (%) | 750 (3.3) |
| EuroSCORE (%) | 442 (2.0) |
| POSSUM (%) | 287 (1.3) |
| P-POSSUM (%) | 1,397 (6.2) |
| Surgery-specific POSSUM (for example, Vasc-POSSUM) (%) | 192 (0.8) |
| Other risk scoring system (%) | 651 (2.9) |

**Abbreviations:** ASA-PS, American Society of Anesthesiologists Physical Status; EuroSCORE, European System for Cardiac Operative Risk Evaluation; POSSUM, Physiology and Operative Severity Score for the enUmeration of Mortality; P-POSSUM, Portsmouth-POSSUM; SORT, Surgical Outcome Risk Tool; SRS, Surgical Risk Scale.

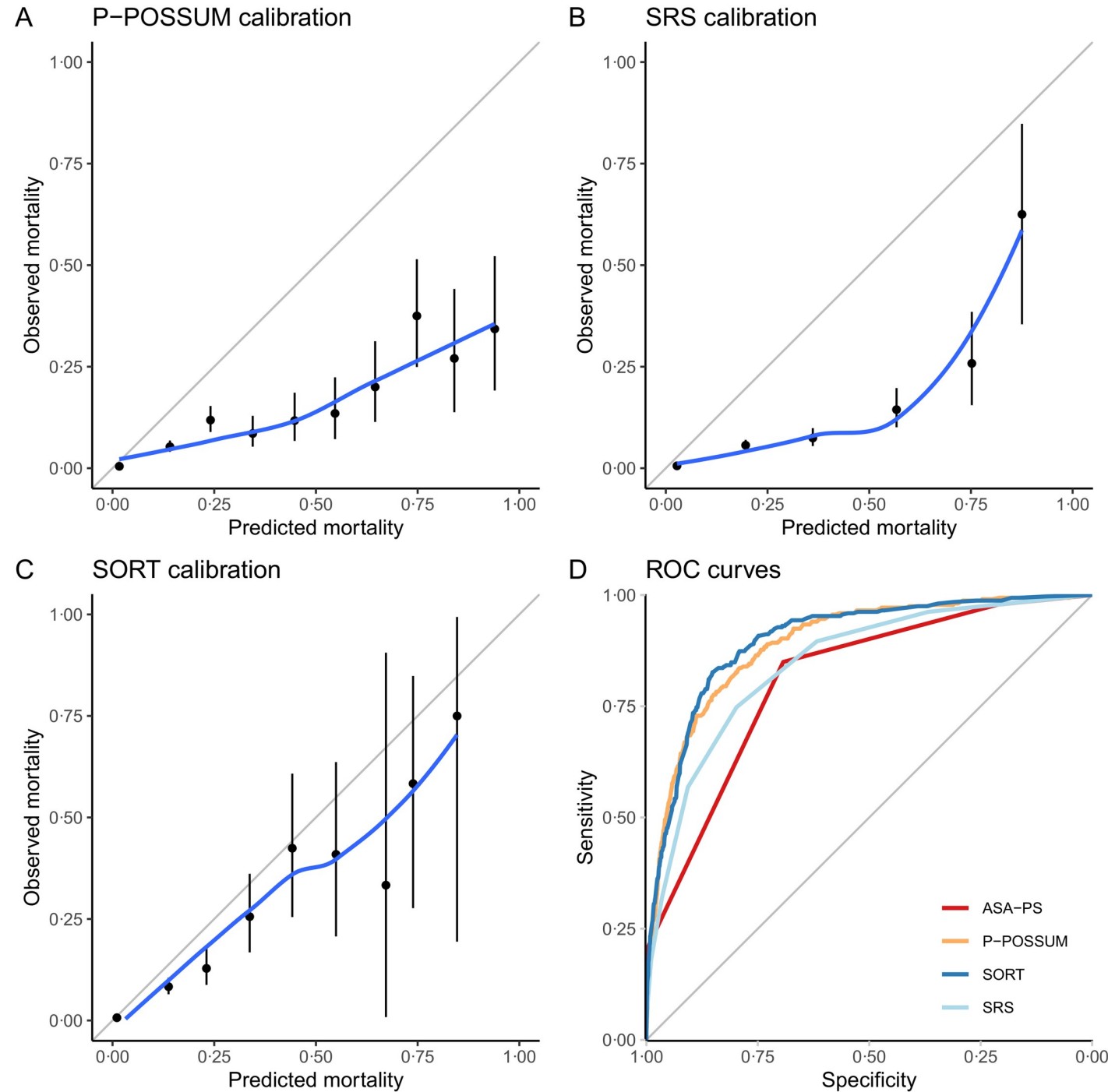

**Fig 2. Calibration plots for the SORT (A), P-POSSUM (B), SRS (C), and ROC curves for the 3 models (D).** In the calibration plots (A–C), nonparametric smoothed best-fit curves (blue) are shown along with the point estimates for predicted versus observed mortality (black dots) and their 95% CIs (black lines) within each decile of predicted mortality. External validation of all 3 models were performed on the entire patient data set (n = 22,631). ASA-PS, American Society of Anesthesiologists Physical Status; CI, confidence interval; P-POSSUM, Portsmouth-Physiology and Operative Severity Score for the enUmeration of Mortality; ROC, Receiver Operating Characteristic; SORT, Surgical Outcome Risk Tool; SRS, Surgical Risk Scale.

## Subjective assessment

There were 188 deaths (1.05%) within 30 days of surgery in the subset of 17,845 patients who had mortality estimates based on clinical judgement and/or ASA-PS alone. Subjective assessment overpredicted risk (Fig 3A, Hosmer–Lemeshow test p < 0.001) but demonstrated good discrimination (Fig 3B and Table 3, AUROC = 0.89, 95% CI: 0.86–0.91), which was not significantly different from the SORT (p = 0.309). Continuous NRI analysis did not show improvement in classification when using the SORT compared with subjective assessment (Table 3 and S4 Text). The 30-day mortality outcomes at each level of clinician risk prediction were cross-tabulated, showing that clinician predictions correlated well with actual mortality outcomes (S2 Table).

## Combining subjective and objective risk assessment

Bootstrapped internal validation yielded an optimism-corrected AUROC of 0.92 for a combined model using both subjective assessment and SORT predictions as independent variables (Table 3); this was better than subjective assessment alone (p < 0.001) and SORT alone (p = 0.021) (Table 4). The model also significantly (p < 0.001) improved reclassification compared with subjective assessment alone in continuous NRI analysis (S4 Text). The improved NRI was largely attributable to the correct downgrading of patient risks—i.e., a large proportion of patients were correctly reclassified as lower risk using the combined model compared with subjective assessment. The DCA also favoured SORT over the other previously published models, but the combined clinician judgement–SORT model again performed best (Fig 4). The effect of combining information from subjective assessment and the SORT is further demonstrated by computing the conditional probabilities of 30-day mortality using the combined model over a full range of predictor values (Fig 5). When assessing the decision curves across all risk thresholds, the combined model outperformed P-POSSUM and SRS, and beyond approximately the 10% risk threshold, P-POSSUM and SRS demonstrated negative net benefits when they were used. The decision curve for our combined model incorporating both subjective assessment and SORT showed increased net benefit across almost the entire range of risk thresholds versus SORT alone.

## Sensitivity analyses

A summary of the different sensitivity analyses is provided in S5 Text. In the first sensitivity analysis (S6 Text), we repeated the main study analyses using the full cohort of patients available from SNAP-2: EPICCS, including those undergoing obstetric procedures, and found that there were minimal differences seen from our main study findings. The SORT was again the best calibrated of the pre-existing models in this larger cohort, and all objective risk tools again overpredicted risk (S1 Fig; Hosmer–Lemeshow p-values all <0.001 for the SORT, P-POSSUM, and SRS). The estimates for AUROC were minimally affected (S1 Fig; AUROC SORT = 0.91, 95% CI: 0.90–0.93; P-POSSUM = 0.90, 95% CI: 0.88–0.92; SRS = 0.85, 95% CI: 0.83–0.88). The AUROC for the SORT was still significantly better than SRS (p < 0.001), but not P-POSSUM (p = 0.121). Subjective assessment in this first sensitivity analysis demonstrated similar overprediction of risk (S2 Fig, Hosmer–Lemeshow test p < 0.001) but similar discrimination (S2 Fig, AUROC = 0.89, 95% CI: 0.87–0.92) to the main study analysis. Differences in discrimination between subjective assessment and SORT were again not significantly different (p = 0.216). Continuous NRI analysis again did not show improvement in classification when using the SORT compared with subjective assessment in this larger group of patients.

For the second sensitivity analysis (S7 Text), we used a previously defined more restrictive inclusion criteria to identify high-risk patients [15, 32]. This yielded a subgroup of 12,985

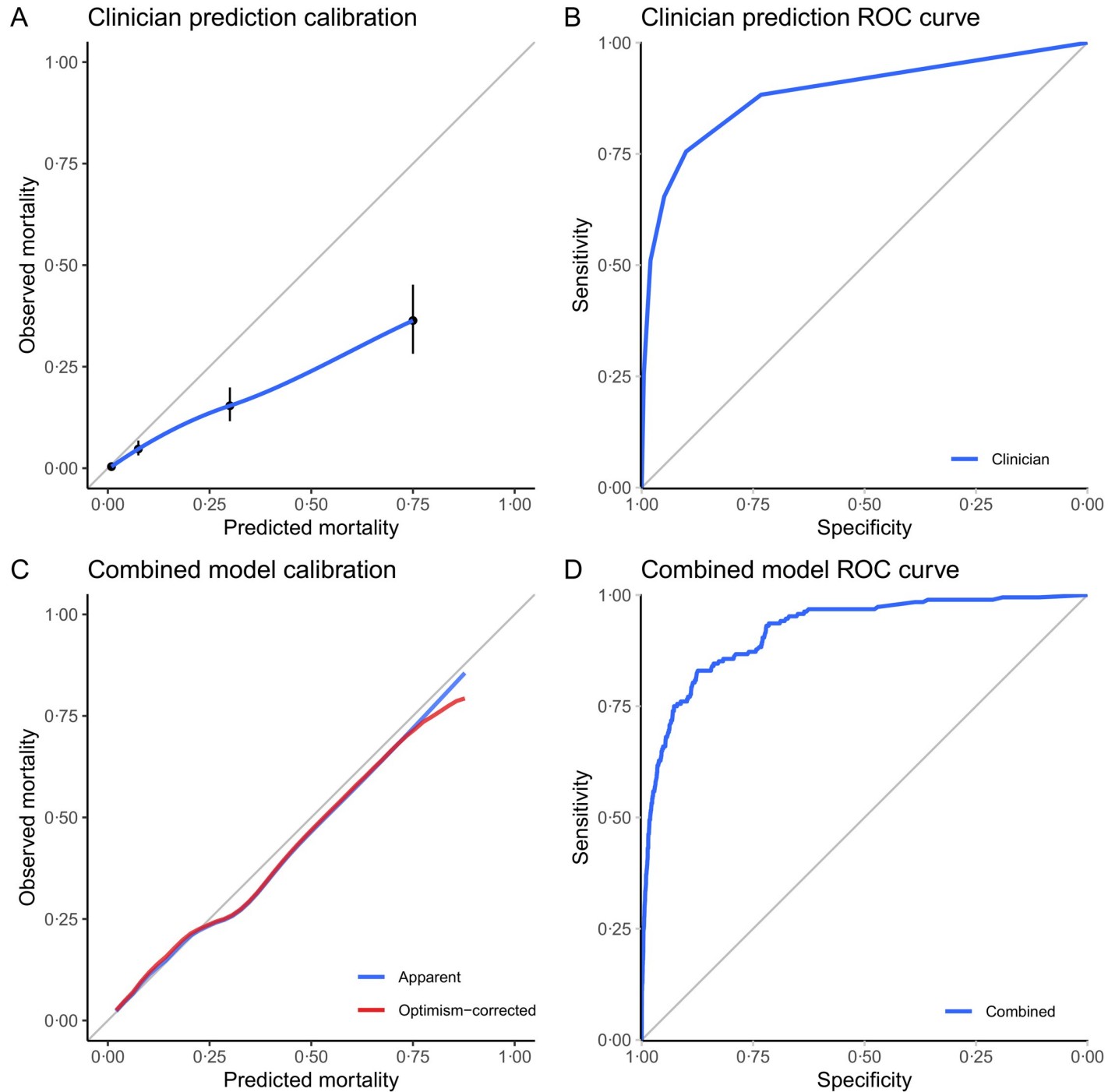

**Fig 3. Calibration plots and ROC curves for subjective clinical assessments (A, B) and the logistic regression model combining clinician and SORT predictions (C, D), validated on the subset of patients in whom clinicians estimated risk based on clinical judgement alone (n = 17,845).** For (A), a nonparametric smoothed best-fit curve (blue) is shown along with the point estimates for predicted versus observed mortality (black dots) and their 95% CIs (black lines) within each range of clinician-predicted mortality. For (C), the apparent (blue) and optimism-corrected (red) nonparametric smoothed calibration curves are shown; the latter was generated from 1,000 bootstrapped resamples of the data set. CI, confidence interval; ROC, Receiver Operating Characteristic; SORT, Surgical Outcome Risk Tool.

**Table 3. Coefficients of the logistic regression model combining subjective clinical assessment with SORT-predicted risk; p-values in a logistic regression model test the null hypothesis that the estimated coefficient is equal to zero using a z-test.**

|  | Coefficient | Standard Error | Z-Statistic | p-Value |
|---|---|---|---|---|
| Intercept | −6.403 | 0.2135 | −30 | <0.001 |
| SORT-predicted risk (per 1% risk) | 0.04028 | 0.007049 | 5.714 | <0.001 |
| **Clinical assessment of risk** | | | | |
| Clinical assessment of risk < 1% | Reference | | | |
| Clinical assessment of risk = 1%–2.5% | 1.487 | 0.2962 | 5.021 | <0.001 |
| Clinical assessment of risk = 2.6%–5% | 2.365 | 0.3177 | 7.444 | <0.001 |
| Clinical assessment of risk = 5.1%–10% | 3.074 | 0.2976 | 10.33 | <0.001 |
| Clinical assessment of risk = 10.1%–50% | 4.156 | 0.2852 | 14.57 | <0.001 |
| Clinical assessment of risk > 50% | 5.028 | 0.3186 | 15.78 | <0.001 |

Abbreviations: SORT, Surgical Outcome Risk Tool.

patients in whom the 30-day mortality rate was 2.01%. In this subgroup, calibrations of P-POSSUM, SRS, and SORT predictions were similar to the full cohort (S3 Fig). The AUROCs were lower in this subgroup (SORT = 0.88, 95% CI: 0.86–0.90; P-POSSUM = 0.86; 95% CI: 0.84–0.89; SRS = 0.81, 95% CI: 0.78–0.84, S3 Fig). The calibration of subjective assessment was again similar to that of the full cohort, and discrimination was reduced but still good (AUROC = 0.85; 95% CI: 0.82–0.89, S3 Fig). The discrimination of subjective assessment in this subgroup was not significantly different from the full cohort (p = 0.155).

The third sensitivity analysis (S8 Text) used the subgroup whose mortality estimate was based on clinical judgement in conjunction with any objective risk tool (n = 4,751, S4 Fig). The AUROC for subjective assessment in this subgroup was 0.88, which was not significantly different from the AUROC in the main cohort (p = 0.769). The calibration of subjective assessment in this subgroup was similar to that in the main cohort, again with a tendency to overpredict risk.

In the fourth sensitivity analysis (S9 Text), we looked for differences in performance of subjective clinical assessment and objective risk tools between the UK and the Australia/New Zealand cohorts (S5 Fig and S3 Table). The 30-day mortality in the Australia/New Zealand cohort (1.09%) was comparable to that of the UK (1.45%, p = 0.127). Visual inspection of calibration plots showed SORT to be worse calibrated in Australasia than the UK. AUROCs for the objective tools in the Australasian subset (P-POSSUM = 0.90, SRS = 0.81, SORT = 0.87) were not significantly different from the AUROCs in the UK subset (P-POSSUM = 0.89, SRS = 0.85,

**Table 4. Performance metrics for clinician prediction versus SORT and versus a logistic regression model combining clinician and SORT prediction.** Calculations based on the subset of patients in whom clinician judgement alone was used to estimate risk (n = 17,845). The reported AUROC for the combined model is the optimism-corrected value from bootstrapped internal validation.

| | ROC | | | Continuous NRI | | |
|---|---|---|---|---|---|---|
| *Model* | *AUROC* | *95% CI* | *p-Value[1]* | *NRI* | *95% CI* | *p-Value[2]* |
| Clinical | 0.886 | 0.858–0.914 | Reference | Reference | | |
| SORT | 0.900 | 0.877–0.923 | 0.309 | 0.073 | −0.062 to 0.208 | 0.288 |
| Combined | 0.920 | 0.899–0.940 | **<0.001** | 0.130 | 0.057–0.202 | **<0.001** |

[1]Differences between AUROCs are tested using DeLong's test for 2 correlated ROC curves with a null hypothesis of no difference.

[2]Differences between continuous NRI statistics are tested using a z-test with a null hypothesis of no difference.

Abbreviations: AUROC, Area Under the Receiver Operating Characteristic curve; CI, confidence interval; NRI, Net Reclassification Improvement.

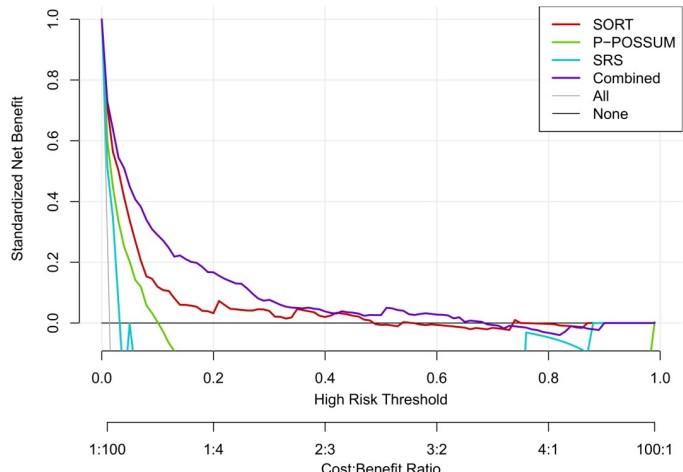

**Fig 4. DCA.** DCA, decision-curve analysis; P-POSSUM, Portsmouth-Physiology and Operative Severity Score for the enUmeration of Mortality; SORT, Surgical Outcome Risk Tool; SRS, Surgical Risk Scale.

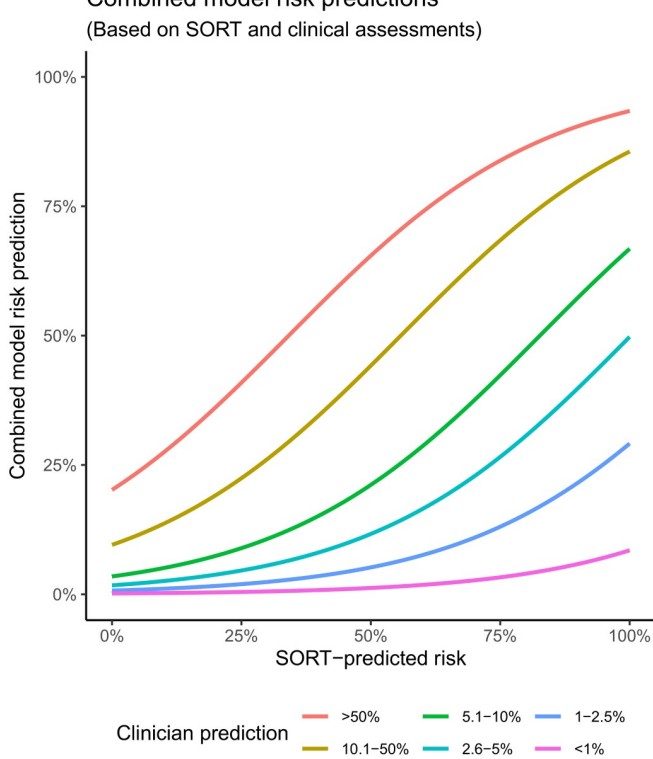

**Fig 5. Predicted risks from combined model, stratified by clinical assessments.** We model the changes to risk predictions (y-axis) based on subjective clinical assessments (coloured lines) as SORT-predicted risks (x-axis) change, to illustrate the change in risk predictions if information from both are combined. P-POSSUM, Portsmouth-Physiology and Operative Severity Score for the enUmeration of Mortality; SORT, Surgical Outcome Risk Tool; SRS, Surgical Risk Scale.

and SORT = 0.90, p > 0.05 for all). The calibration of subjective clinical assessment was comparable in the 2 geographical subgroups, and there were also no significant differences in AUROCs (Australasia: 0.88, UK: 0.89, p = 0.860, S6 Fig).

For the fifth sensitivity analysis (S10 Text), we used the subgroup of patients who had no missing P-POSSUM variables (n = 18,362; see S1 Table for patient characteristics). Patients with complete P-POSSUM variables appeared to be older, have higher ASA-PS grades, and undergo higher-severity surgery in comparison with those with missing P-POSSUM variables. The AUROC for clinical assessments in the subgroup with full P-POSSUM variables was 0.90, which was not significantly different from the AUROC obtained for clinical assessments in the main study analysis (p = 0.587), and the predictions were similarly calibrated to clinical assessments in the main study analysis, again with a tendency to overpredict risk (S7 Fig). When comparing the performance of P-POSSUM (AUROC = 0.89), SRS (AUROC = 0.84), and SORT (AUROC = 0.90) in this subgroup, the performance was again similar to that of the main study cohort (p > 0.05 for all comparisons).

In the sixth and final sensitivity analysis (S11 Text), we evaluated the AUROC and calibration of the SORT–clinical judgement model in subgroups of patients according to surgical specialty (S4 Table). We found that the AUROC remained high within these subgroups (ranging from 0.87, 95% CI 0.75–0.98 in 1,033 cardiothoracic surgical patients through to 0.95, 95% CI 0.90–0.99 in 4,309 gynaecology and urology patients). Calibration was also good across different specialties, with the exception of vascular surgery (674 patients, AUROC 0.88, 95% CI 0.82–0.94; Hosmer-Lemeshow p-value = 0.009).

## Discussion

We present data from an international cohort of patients undergoing inpatient surgery with a low risk of recruitment bias. Despite a plethora of options for objective risk assessment, in over 80% of patients, subjective assessment alone was used to predict 30-day mortality risk. All previously published risk models were poorly calibrated for this cohort of patients, reflecting the common problem of calibration drift over time. However, the combination of subjective clinical assessment with the parsimonious SORT model provides an accurate prediction of 30-day mortality, which is significantly better than any of the methods we evaluated used on their own. These findings should give confidence to clinicians that the combined SORT–clinical judgement model can be used to support the appropriate allocation of finite resources and to inform discussions with patients about the risks of surgery. The combined model accurately downgraded predicted risk compared with other methods; therefore, application of this approach may result in fewer low-risk patients inappropriately admitted to critical care (thus easing system pressures) and may result in fewer patients having their surgery cancelled for the lack of a critical care bed [7]. Finally, application of the SORT–clinical judgement model may assist hospital managers and policy makers in determining the likely demand for postoperative critical care, thus supporting best practice at the hospital, regional, or national level. This new model will now be incorporated into an open-access risk-assessment system (http://www.sortsurgery.com/), enabling clinicians to combine their clinical estimation of risk and the SORT model to evaluate patient risk from major surgery.

To our knowledge, this is the first study comparing subjective and objective assessment for predicting perioperative mortality risk in a large multicentre international cohort. The highest-quality previous studies in this field have been challenged by recruitment bias because of the predominant participation of research active centres and the need for patient consent. For example, the METS (Measurement of Exercise Tolerance before Surgery) study ([15], which compared clinical assessment of functional capacity with exercise testing, self-assessment, and

a serum biomarker in 1,401 patients, and the VISION study (Vascular Events in non-cardiac surgery cohort study) [32], which evaluated postoperative biomarkers in 15,133 patients, had 27% and 68% screening to recruitment rates, respectively. One way of overcoming such biases would be to study the accuracy of prognostic models using routinely collected or administrative data; however, this is unlikely to enable the evaluation of subjective assessments in multiple centres. Our study avoided these issues through prospective data collection in an unselected cohort with an ethical waiver for patient consent. The mortality in our sample closely matches that recorded in UK administrative data of patients undergoing major or complex surgery [11], therefore supporting our assertion that our cohort was representative of the 'real-world' perioperative population.

Our observation that the majority of risk assessments conducted for perioperative patients do not involve objective measures is also noteworthy because subjective assessment is currently almost never incorporated into risk prediction tools for surgery. One exception is the American College of Surgeons National Surgical Quality Improvement Program Surgical Risk Calculator [33], which incorporates a 3-point scale of clinically assessed surgical risk (normal, high, or very high) to supplement a calculated prediction of mortality and various short-term outcomes. However, this system is proprietary, has rarely been evaluated outside the US, and is substantially more complex than the SORT–clinical judgement model, with 21 input variables compared with 8. Furthermore, their methodology for developing this 'uplift' was quite different from ours, using a panel of 80 surgeons to evaluate 10 case scenarios and grade them in retrospect.

We recognise some limitations to our study. First, models predicting rare events may appear optimistically accurate, as a model that identifies every patient as being at low risk of mortality in a group in which the probability of death approaches 0% would almost always appear to be correct. For this reason, we undertook several sensitivity analyses, including one that evaluated the performance of the various risk-assessment methods in a subgroup of patients who have been defined as high risk in previous studies of prognostic indicators and in whom the mortality rate was higher. We found that the performance of the SORT and subjective assessment remained good and compared favourably with previous evaluations of more complex risk-assessment methods [15,32]. Second, whilst we assumed that subjective assessments were truly clinically based judgements, because this was a pragmatic unblinded study, it was possible that information from other sources may have subconsciously influenced these assessments. For this reason, we undertook the second sensitivity analysis, which refuted this possible risk. Third, the very act of estimating mortality risk may lead clinicians to take actions that improve that risk, therefore biasing the outcome of the assessments made and in particular affecting the calibration of subjective risk estimates. The only way to avoid this risk would be to have used subjective assessments made by clinicians independent of the clinical management of individual patients, and this may be an interesting opportunity for future research. Fourth, since we undertook this study, other promising risk-assessment methods have been developed, including the Combined Assessment of Risk Encountered in Surgery (CARES) system, which was developed using electronic health records; unfortunately, we were unable to externally validate this system because we did not collect all the required variables [34]. We also did not evaluate the accuracy of other risk prediction methods such as frailty assessment or cardiopulmonary exercise testing. However, this was not an a priori objective of our study [18]; furthermore, our observation of the lack of 'real-world' use of these types of predictors is in itself an important finding, particularly given the substantial interest in such measures (some of which carry considerable cost) in the research literature [15,35]. Fifth, the UK cohort was substantially larger than the Australasian cohort; however, we found no significant differences in mortality or accuracy of the various risk-assessment methods between the 2

geographical groups. Finally, the study was conducted entirely in high-income countries; therefore, our findings should now be tested in low- and middle-income nations in order to evaluate global generalisability.

Our finding that the combination of subjective and SORT-based assessment is the best approach is important because it is likely to have face validity with clinicians, thereby improving the likelihood that our new model will be incorporated into clinical practice. There is a sound rationale for this finding, as it is likely that clinicians consider otherwise unmeasured factors that they recognise as important, such as severity of comorbid diseases, frailty, socioeconomic status, patient motivation, and anticipated technical challenges. Modern approaches to risk assessment using machine learning [36] provide promise for automation of risk prediction and incorporating data and calculations that clinicians may subconsciously consider when making subjective decisions; however, even these methods do not substantially outperform our simpler approach and are currently limited by recruitment biases and lack of availability. Future research could evaluate the benefits of incorporating clinical judgement into risk-assessment methods in medicine more generally.

Implementation of a widely available, parsimonious, and free-to-use risk-assessment tool to guide clinical decision-making about critical care allocations and other aspects of perioperative care may now be considered particularly important in view of the likely prevalence of endemic COVID-19 leading to an increased demand for critical care facilities. Therefore, now more than ever, risk-based allocation of these resources is important for the benefit of individual patients and the hospitalised population as a whole. Further to this, application of either the SORT or the SORT–clinical judgement model to perioperative population data may assist healthcare policy makers and managers in modelling the likely demand for postoperative critical care, thus improving system level planning and resource utilisation. Based on the results of this large generalisable cohort study, the focus of the perioperative academic community could now shift from evaluation of which risk prediction method might be best to testing the impact of SORT–clinical judgement-based decision-making on perioperative outcomes.

In conclusion, the combination of subjective and objective risk assessment using the SORT calculator provides a more accurate estimate of 30-day postoperative mortality than subjective assessment alone. Implementation of the SORT–clinical judgement model should lead to better clinical decision-making and improved allocation of resources such as critical care beds to patients who are most likely to benefit.

## Supporting information

**S1 Text. STROBE checklist.** STROBE, Strengthening the Reporting of Observational Studies in Epidemiology.
(PDF)

**S2 Text. TRIPOD checklist.** TRIPOD, Transparent Reporting of a multivariable prediction model for Individual Prognosis Or Diagnosis.
(DOCX)

**S3 Text. Case report form.**
(DOCX)

**S4 Text. Continuous net reclassification index analysis and reclassification tables.**
(DOCX)

**S5 Text. Sensitivity analyses overview.**
(DOCX)

**S6 Text. Sensitivity analysis 1.**
(DOCX)

**S7 Text. Sensitivity analysis 2.**
(DOCX)

**S8 Text. Sensitivity analysis 3.**
(DOCX)

**S9 Text. Sensitivity analysis 4.**
(DOCX)

**S10 Text. Sensitivity analysis 5.**
(DOCX)

**S11 Text. Sensitivity analysis 6.**
(DOCX)

**S12 Text. Acknowledgments and full list of SNAP2: EPICCS collaborators.** SNAP2:
EPICCS, Second Sprint National Anaesthesia Project: EPIdemiology of Critical Care provision
after Surgery.
(DOCX)

**S1 Table. Characteristics of the patient subgroups used in all sensitivity analyses.** ASA-PS,
American Society of Anesthesiologists Physical Status; COPD, Chronic Obstructive Pulmo-
nary Disease; IQR, interquartile range; P-POSSUM, Portsmouth-Physiology and Operative
Severity Score for the enUmeration of Mortality; SORT, Surgical Outcome Risk Tool; SRS,
Surgical Risk Scale.
(DOCX)

**S2 Table. Confusion matrix of patients 30-day mortality outcomes versus clinician predic-
tions.** (%) represents row percentage.
(DOCX)

**S3 Table. AUROCs of the objective risk tools and subjective assessment, compared
between the UK and Australian/New Zealand data subsets.** We found no significant differ-
ence in discrimination using any of the risk prediction tools or using subjective assessment
when comparing their performance in the UK and Australian/New Zealand data sets.
AUROC, Area Under Receiver Operating Characteristic curve; P-POSSUM, Portsmouth-
Physiology and Operative Severity Score for the enUmeration of Mortality; SORT, Surgical
Outcome Risk Tool; SRS, Surgical Risk Scale.
(DOCX)

**S4 Table. Discrimination and calibration performance of the new combined prediction
model in different specialty subgroups.**
(DOCX)

**S1 Fig. Calibration plots for the SORT (A), P-POSSUM (B), SRS (C), and ROC curves for
the 3 models (D) validated in the whole patient cohort, including those undergoing obstet-
ric procedures.** In the calibration plots (A–C), nonparametric smoothed best-fit curves (blue)
are shown along with the point estimates for predicted versus observed mortality (black dots)
and their 95% CIs (black lines) within each decile of predicted mortality. External validation of
all 3 models were performed on the entire SNAP-2: EPICCS patient data set (n = 25,854). CI,
confidence interval; P-POSSUM, Portsmouth-Physiology and Operative Severity Score for the

enUmeration of Mortality; ROC, Receiver Operating Characteristic; SNAP-2: EPICCS, Second Sprint National Anaesthesia Project: EPIdemiology of Critical Care provision after Surgery; SORT, Surgical Outcome Risk Tool; SRS, Surgical Risk Scale.
(PDF)

**S2 Fig. Calibration plots and ROC curves for subjective clinical assessments (A, B) and the logistic regression model combining clinician and SORT predictions (C, D), validated on the subset of patients in whom clinicians estimated risk based on clinical judgement alone, drawn from the full SNAP-2: EPICCS data set, including patients who underwent obstetric surgery (n = 21,325).** For (A), a nonparametric smoothed best-fit curve (blue) is shown along with the point estimates for predicted versus observed mortality (black dots) and their 95% CIs (black lines) within each range of clinician predicted mortality. For (C), the apparent (blue) and optimism-corrected (red) nonparametric smoothed calibration curves are shown, the latter was generated from 1,000 bootstrapped resamples of the data set. CI, confidence interval; ROC, Receiver Operating Characteristic; SNAP-2: EPICCS, Second Sprint National Anaesthesia Project: EPIdemiology of Critical Care provision after Surgery; SORT, Surgical Outcome Risk Tool.
(PDF)

**S3 Fig. Calibration plots for SORT (A), P-POSSUM (B), SRS (C), and clinical assessments (E) and ROC curves for the 3 models (D) and clinical assessments (F), validated in the sensitivity analysis patient subset with restricted inclusion criteria (n = 12,985).** The AUROCs for P-POSSUM, SRS, SORT, and clinical assessments were 0.863, 0.810, 0.875, and 0.853 in this subgroup, respectively. AUROC, Area Under Receiver Operating Characteristic curve; P-POSSUM, Portsmouth-Physiology and Operative Severity Score for the enUmeration of Mortality; SORT, Surgical Outcome Risk Tool; SRS, Surgical Risk Scale.
(PDF)

**S4 Fig. Calibration plot (A) and ROC curve (B) for clinical assessments, validated in the sensitivity analysis patient subgroup in which clinical assessments were made in conjunction with 1 or more other risk prediction tools (n = 4,786).** The AUROC for clinical assessments was 0.880 in this subgroup. AUROC, Area Under Receiver Operating Characteristic curve.
(PDF)

**S5 Fig. Calibration plots (A to F) and ROC curves (G & H) for objective risk tools, validated in patients stratified by their country groups.** There was minimal difference between countries. ROC, Receiver Operating Characteristic.
(PDF)

**S6 Fig. Calibration plots (A & B) and ROC curves (C & D) for clinical assessments, validated in patients stratified by their country groups.** There was minimal difference between countries. ROC, Receiver Operating characteristic Curve.
(PDF)

**S7 Fig. Calibration plots for SORT (A), P-POSSUM (B), SRS (C), and clinical assessments (E) and ROC curves for the 3 models (D) and clinical assessments (F), validated in the sensitivity analysis patient subset with complete P-POSSUM variables (n = 18,362).** The AUROCs for P-POSSUM, SRS, SORT, and clinical assessments were 0.893, 0.838, 0.899, and 0.896 in this subgroup, respectively. AUROC, Area Under Receiver Operating Characteristic curve; P-POSSUM, Portsmouth-Physiology and Operative Severity Score for the enUmeration of

Mortality; SORT, Surgical Outcome Risk Tool; SRS, Surgical Risk Scale.
(PDF)

**S1 Data. Raw data.**
(ZIP)

## Author Contributions

**Conceptualization:** S. Ramani Moonesinghe.

**Data curation:** Danny J. N. Wong, Andrew M. Wilson, Scott Popham, Lisa M. Barneto, S. Ramani Moonesinghe.

**Formal analysis:** Danny J. N. Wong, Steve Harris, Arun Sahni, James R. Bedford, S. Ramani Moonesinghe.

**Funding acquisition:** Steve Harris, Richard Shawyer, S. Ramani Moonesinghe.

**Investigation:** Danny J. N. Wong, Steve Harris, Arun Sahni, James R. Bedford, Andrew M. Wilson, Helen A. Lindsay, Lisa M. Barneto, Paul S. Myles, S. Ramani Moonesinghe.

**Methodology:** Danny J. N. Wong, Steve Harris, Laura Cortes, Richard Shawyer, Paul S. Myles, S. Ramani Moonesinghe.

**Project administration:** Danny J. N. Wong, Steve Harris, Laura Cortes, Andrew M. Wilson, Helen A. Lindsay, Doug Campbell, Scott Popham, Lisa M. Barneto, Paul S. Myles, S. Ramani Moonesinghe.

**Resources:** Danny J. N. Wong, Steve Harris, Arun Sahni, James R. Bedford, Laura Cortes, Helen A. Lindsay, Doug Campbell, Scott Popham, Lisa M. Barneto, Paul S. Myles, S. Ramani Moonesinghe.

**Software:** Danny J. N. Wong.

**Supervision:** Steve Harris, Doug Campbell, Paul S. Myles, S. Ramani Moonesinghe.

**Validation:** Danny J. N. Wong, Steve Harris, S. Ramani Moonesinghe.

**Visualization:** Danny J. N. Wong, Steve Harris, S. Ramani Moonesinghe.

**Writing – original draft:** Danny J. N. Wong, Steve Harris, S. Ramani Moonesinghe.

**Writing – review & editing:** Danny J. N. Wong, Steve Harris, Arun Sahni, James R. Bedford, Laura Cortes, Richard Shawyer, Andrew M. Wilson, Helen A. Lindsay, Doug Campbell, Scott Popham, Lisa M. Barneto, Paul S. Myles, S. Ramani Moonesinghe.

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
