## [Decision Letter · Decision Letter 0]

10 Mar 2020

Dear Dr. Moonesinghe,

Thank you very much for submitting your manuscript "Comparing subjective and objective risk assessment for predicting mortality after major surgery - an international prospective cohort study" (PMEDICINE-D-19-04548) for consideration at PLOS Medicine. 

Your paper was evaluated by a senior editor and discussed among the editors here. It was also discussed with an academic editor with relevant expertise, and sent to independent reviewers, including a statistical reviewer. The reviews are appended at the bottom of this email and any accompanying reviewer attachments can be seen via the link below:

[LINK]

In light of these reviews, I am afraid that we will not be able to accept the manuscript for publication in the journal in its current form, but we would like to consider a revised version that addresses the reviewers' and editors' comments. Obviously we cannot make any decision about publication until we have seen the revised manuscript and your response, and we plan to seek re-review by one or more of the reviewers. 

We expect to receive your revised manuscript by Mar 31 2020 11:59PM. Please email us (plosmedicine@plos.org) if you have any questions or concerns.

We look forward to receiving your revised manuscript. 

Sincerely,

Louise Gaynor-Brook, MBBS PhD

Associate Editor 

PLOS Medicine

plosmedicine.org

General comment: Please cite reference numbers in square brackets, leaving a space before the reference bracket, and removing spaces between reference numbers where more than one reference is cited e.g. '... postoperative complications [5,6].'

General comment: Please refer to a more specific part of your appendix to make clear to readers to what you are referring e.g. Figure S1, Table S1, etc.

General comment: Throughout the ms, please quote exact p values or "p<0.001", unless there is a specific statistical justification for quoting smaller exact numbers. 

General comment: Please add line numbers to your revised manuscript.

Data Availability Statement: PLOS Medicine requires that the de-identified data underlying the specific results in a published article be made available, without restrictions on access, in a public repository or as Supporting Information at the time of article publication, provided it is legal and ethical to do so. Please provide the URL and any other details that will be needed to access data stored in the public repository. 

Title: Please revise your title according to PLOS Medicine's style, placing the study design in the subtitle (ie, after a colon). We suggest “Subjective and objective risk assessment for predicting mortality

after major surgery: an international prospective cohort study” or similar

Abstract Background: Please expand upon the context of why the study is important. The final sentence should clearly state the study question, and clarify the aims of the study e.g. to compare the accuracy of freely-available objective surgical risk tools with subjective clinical assessment in predicting 30-day mortality

Please combine the Methods and Findings components of your Abstract into one subsection with the heading Methods and Findings’

Please include the study design, brief demographic details of the population studied (e.g. age, sex, types of surgery, etc) and further details of the study setting (e.g. what type of hospitals - secondary / tertiary care? Rural / urban? etc.), and main outcome measures.

In your abstract and elsewhere, please quote p values alongside 95% CI where available. 

Please revise the sentence beginning ‘We included consecutive adults…’ to clarify. Please revise 'consecutive adults’ and ‘undergoing inpatient surgery for one week’

In the last sentence of the Abstract Methods and Findings section, please describe the main limitation(s) of the study's methodology.

Please begin your Abstract Conclusions with "In this study, we observed ..." or similar. Please address the study implications, emphasising what is new without overstating your conclusions.

Please tone down subjective language such as ‘highly accurate’ 

Please avoid vague statements such as " Clinicians can use this….", to instead highlight that ‘This may be of value in helping to stratify… ‘

At this stage, we ask that you include a short, non-technical Author Summary of your research to make findings accessible to a wide audience that includes both scientists and non-scientists. The Author Summary should immediately follow the Abstract in your revised manuscript. This text is subject to editorial change and should use non-identical language distinct from the scientific abstract. Please see our author guidelines for more information: https://journals.plos.org/plosmedicine/s/revising-your-manuscript#loc-author-summary Please ensure to include a final bullet point under ‘What do these findings mean?’ to describe the main limitation(s) of the study.

Introduction 

Please explain the need for and potential importance of your study. Indicate whether your study is novel and how you determined that. 

Please define QI

Methods

Please include the completed STROBE / TRIPOD checklists as Supporting Information and refer to the relevant supplementary files in your Methods section. When completing the checklists, please use section and paragraph numbers, rather than page numbers.

Please adapt "STROBE diagram" to "participant flowchart", or similar.

Did your study have a prospective protocol or analysis plan? Please state this (either way) early in the Methods section. If a prospective analysis plan was used in designing the study, please include the relevant prospectively written document with your revised manuscript as a Supporting Information file to be published alongside your study, and cite it in the Methods section. If no such document exists, please make sure that the Methods section transparently describes when analyses were planned, when/why any data-driven changes to analyses took place, and what those changes were. 

Please provide more detail on the hospitals included in the study e.g. secondary / tertiary care, how many from each country, etc.

Please provide detail on the inclusion criteria, as this is notably missing from your supplementary material. 

Please provide further detail on the institutional research and development department approvals for Northern Ireland, Australia and New Zealand as you have for the UK. This should include the names of the institutional review board(s) that provided ethical approval.

Results

Please mention in the main text of your Results that 4,891 surgical episodes were excluded from analysis (as indicated in your STROBE diagram)

Please quote p values alongside 95% CI where available. 

Please refer to supplementary figures for the results presented for the sensitivity analyses.

Please consider whether additional sensitivity analyses could be performed to at least partially address comments about inclusion of certain specialties that might bias the overall findings in a particular direction. Please ensure that it is made clear in your Methods that any such additional analyses are non-prespecified. 

Table 1 - please revise ‘Xmajor’. 

Tables 3 & 4 - please define all abbreviations used in the table legend. When a p value is given, please specify the statistical test used to determine it in the table legend.

Tables S3 & S4 - please define all abbreviations used in the table legend.

Figure 2 - please define all abbreviations used in the figure legend

Figure 4 - please provide a figure legend

Supplementary Figures 3 and 4 - please provide a figure legend

Discussion 

Please present and organize the Discussion as follows: a short, clear summary of the article's findings; what the study adds to existing research and where and why the results may differ from previous research; strengths and limitations of the study; implications and next steps for research, clinical practice, and/or public policy; one-paragraph conclusion.

Please tone down subjective language such as ‘highly accurate’ 

Please avoid vague statements such as " Clinicians can use this….", to instead highlight that ‘This may be of value in helping to stratify… ‘

References

Please provide names of the first 6 co-authors for each paper referenced, before ‘et al’

Noting reference 11, please ensure that all references have full access details. 

Supplementary Files

Please clearly label each supplementary file / table / figure with a distinct name and ensure that a reference is made to each component in the main text of the manuscript. 

Please ensure that all components of your supplementary files e.g. inclusion criteria are included in your resubmitted manuscript.

Comments from the reviewers:

Reviewer #1: "Comparing subjective and objective risk assessment for predicting mortality after major surgery - an international prospective cohort study of 26,216 patients" evaluates three publicly-available objective risk tools against clinicians' subjective judgment on mortality prediction, and further integrates the best-performing objective risk tool with said subjective judgment to produce an even better-performing logistic regression model.

The prospective nature of the study, diverse demographics involved (over 26,000 patients from 274 hospitals in the United Kingdom, Australia and New Zealand) and open availability of the examined objective risk tooks (P-POSSUM, SRS and SORT) are particular strengths of the manuscript. Overall, the work seems to have the potential to broadly benefit surgical management practice. However, there are a number of points that might be expanded upon.

Firstly, on the comparison methods; the authors may wish to clarify their definition of objective vs. subjective for risk assessment procedures. In particular, the "objective" Surgical Risk Scale (SRS) appears to comprise a summation of ASA-PS scores together with the CEPOD and BUPA scores ("The surgical risk scale as an improved tool for risk-adjusted analysis in comparative surgical audit", Sutton et al., 2002), but the ASA-PS is itself considered "subjective" assessment (Page 8). Moreover, it is not clear whether CEPOD and BUPA are any less "subjective" than ASA-PS, from their descriptions.

Also, for P-POSSUM, it is noted that a number of biochemical/haematological parameters were missing in practice ("Missing data" section, Page 9), and in these cases, normal data was assumed/imputed. Moreover, there remained a small number of cases with further missing data beyond these bio/haemo parameters (as understood from "Following imputation, we performed a complete case analysis as we considered the proportion of cases with missing data in *the remaining variables* to be low [1.08%]"). This practice however seems problematic, in that close to half of the variables used for P-POSSUM appear to be biochemical/haemotological (i.e. haemoglobin, WBC, urea, sodium, potassium). Simply assuming normal values for these variables would then appear to rob P-POSSUM of quite a bit of its utility, since these assumptions constitute unsubstantiated evidence towards better patient outcomes. The authors might clarify as to exactly how many cases were affected by missing data for P-POSSUM, and consider an analysis of only those cases where full data was available for P-POSSUM, for a fair comparison.

In general, the authors might consider summarizing the parameters & criteria used for the various objective methods as supplementary material.

Secondly, on the use of AUROC as a main quantative assessment metric of the various methods - it is stated that the 30-day risk of death was predicted as one of six categorical responses, for all methods used (Page 7). Then, although the continuous net reclassification improvement statistic (NRI) is cited (Page 8), the authors might clarify in greater detail as to how this relates to the construction of ROC curves (i.e. Figure 2 & 3). For example, if the prediction is 0.5% and the patient indeed survives for 30 days, what sensitivity/specificity does this amount to as opposed to if the patient had not survived (we assume a binary outcome)? While readers might be familiar with the usual binary ROC formulation, the implementation of multiple categories might warrant more description.

Further on the categorial prediction, the calibration graphs in Figures 2 and 3 appear to suggest that the various methods output different numbers of point estimates. For example, P-POSSUM has 10, SRS has 6, SORT has 9 and clinicians have 4 (as opposed to the six categories implied in the Dataset section). While it seems that a greater number of point estimates improves the level of detail of the corresponding ROC curve, the effect seems exaggerated for SORT and SRS; in particular, the ASA-PS ROC curve in Figure 2 seems to be a piecewise construction from about 3 datapoints, while the SRS curve likewise seems to be piecewise constructed from about 6 datapoints. However, the SORT and SRS curves have much finer detail (i.e. have an independent value for each 0.01 change in specificity or less), despite only having slightly more point estimates than SRS. The authors might wish to explain this discrepancy.

Thirdly, while there is detailed analysis of the aggregate statistics for the various methods (including on a high-risk patient subset), an additional inter-model analysis at finer granularity would seem to be appropriate. In particular, do the various objective and subjective models tend to agree/disagree on specific patients, and in cases where they strongly disagree (i.e. one method predicts a very low mortality risk, while another method predicts a significant risk, for the same patient), what are the factors that might have led to this disagreement? Such an analysis would help to determine whether the choice of particular models (e.g. the combined logistic regression model vs. SORT alone) is strictly beneficial for all patients (strictly dominating method), or if it may shift the risks of inaccurate prediction from one group of patients onto another group.

A confusion matrix of the six prediction categories vs. binary outcomes at a reasonable ROC operating point for each method would also be illuminating.

Fourthly, the authors may wish to comment further on the practical implications of accurate mortality prediction (it is noted in the Discussion section that accurate prediction may result in fewer inappropriate admissions to critical case, though there remains a lack of "real-world" clinical uptake); will such predictions be used by clinicians/patients in deciding whether to commence surgery? Indeed, given that mortality estimates were made by the perioperative team beforehand for participating patients, did these estimates have any impact on the treatment offered (i.e. the abovementioned reduction in inappropriate admissions)?

Finally, while perhaps somewhat out of the scope of this manuscript, the authors might consider exploring data-centric machine learning methods in the future (i.e. train models directly from the available patient demographic features)

Reviewer #2: In this international prospective cohort study, the authors adress the very important issue of preopeative risk assessment for mortality. 

The article is clear an well written, but there is an important methodological question that have to be considered.

1) It is certainly a very good point to test the subjetive risk assesment. However a main limitation is that the physicians could not have used objective assessments scores to help them answering the question proposed. These cases should have been excluded for all analysis initially (flowchart 1). Additionally, patients for whom ASA score was used were kept in the other analysis and classified as belonging to the `subjective analysis `group. Altough ASA score is the oldest and has a lot of subjectivity in itself, it could not be considered that no score was used. Also, the numbers in Flowchart 1 and Table 2 are a bit confusing (23540 patients with clinical judgment only and 9928 with ASA score in Table 2 and in flowchart 21325 patients inluded). The main analysis should have been done only with patients classified by the clinical judgment according to the answer of the question proposed and not using any additinal tool, including ASA.

2) It is stated in methods that data imputing was done to the laboratory values that were missing to the POSSUM score. Please provide how many imputation was performed. Additionaly, assuming that the values are normal in patients undergoing elective surgery without a blood withdraw available, that may not be accurate for emergency patients, for whom there was no time to perform a blood test. 

3) Although there is no consensus of what is the best risk stratification tool, in the discussion the authors mentioned that the limitation of the SQIP risk calculator was never validated outside the US. This would have been a good oportunity to do so because, although it involves more variables, it is online available for free and could be easily implemented in an electronic chart, for exemple. 

Reviewer #3: Dear authors

Thank you to provide me the opportunity to read your manuscript, it reports a comparison of objective and subjective predictive tools to predict postoperative mortality after major surgery.

While the objectives of the study are interesting and would have some potentially important clinical applications, several major limitations limit the interpretation of the observed results.

My key suggestions to improve the impact of this study are:

To select preoperative (not P-POSSOM) scores that do not already include a subjective component (neither SORT nor SRS).

To not attend to create a universal predictive tool working for any type of surgeries.

To better describe the subjective score introduced: inter-rater variability, distributions and eventually to reduce the number of strata to limit variability.

To emphasize the results provided by the decision curves rather than focussing on AUC ROC in models with poor calibration.

I summarized the most important methodological concerns below

1 - Population: The study population includes a non-representative mix of cases including some surgical specialties that are associated with high postoperative mortality and in which general predictive tools don't work well (i.e cardiac surgery - 3.9% of the cohort, 5.7% of the observed deaths).

The objective of the study is to evaluate the predictive performances of scores already described elsewhere. Therefore we are not seeing representative samples and we are simply evaluating the performances of some predictive models. Unfortunately, the combination of the clinical prediction and the other scores is dependent of the study population case-mix, because a new model has been fit. Whether this combined model is working on other populations with a different case-mix is unknown and must be discussed properly as an important limitation.

2 - Population (2): The inclusion of obstetric is a usual mistake we can find in similar studies. It increased the number of patients in the cohort, but it is not associated with any deaths. Therefore, it dropped the average apparent mortality (no deaths after obstetrics in the study we are discussing today) and make the methodological tools used to compare the predictive models not accurate and potentially biased. Further, with no deaths and a clearly identified subgroup (which is included in the calculation of the scores), the inclusion of OB patients artificially increases discrimination performances but is not clinically relevant. Which clinician(s) would use the same score to predict accurately the outcome of OB and cardiac patients?

3 - Major methodological mistake: 

We are dealing with low-frequency primary outcome (i.e. 30d mortality = 1.2% - 317 deaths in 26,616 patients).

P-POSSUM, SRS and SORT showed terrible calibrations in figure 2, where we are observing dramatic overestimations of the risk of death (The calibration curves are way down in the lower right part of the calibration plots). This common methodological concern is poorly detected with ROC curves. Combining infrequent outcomes and poorly calibrated predictive models constantly produce very high AUC while models have no clinical values. That is what we observed here - and in many other studies dealing with similar population characteristics. including OB patients amplified the phenomenon.

4 - Major methodological mistake (2):

SORT : Truly preoperative, include ASA (subjective and wide inter-rater variability), fine description of the surgical procedures

P-POSSUM: More objective risk score with less inter-rater variability, but includes intraoperative variable not available SRSpreoperatively

SRS: Truly preoperative, include ASA (subjective and wide inter-rater variability) and a subjective stratification associated with the planned surgery.

The new Subjective assessment introduced in this study: Preoperative, very subjective, 6 categories, somewhat like ASA. NO quantification of the inter-rater variability which is suspected to be very high in the not extreme categories.

2 of the 3 scores evaluated already include a clinical subjective evaluation of the preoperative patients' characteristics (i.e. ASA with all the limitations we know)

1 score includes intraoperative characteristics which make the comparison impossible.

The authors finally combined SORT and the clinical prediction they introduced. They demonstrated in the study population (after model fitting) that the predictive performance are somewhat better, and that the calibration is way better (they fit a new model...)

SORT already includes ASA, how does ASA interacted with the clinical prediction which look very similar.

While it is recognized that the inter-rater variability of ASA is huge, what do we know about the inter-rater variability of the clinical prediction described here?

5 - The use of decision curves is a very strong methodological part of this study, Unfortunately, their description and interpretation are somewhat neglected compared to useless ROC comparison. The reviewer strongly recommend to expand the description , interpretation and disccusion of the decision curves which actuaaly provide the information that clinicians are seeking.

Some specific comments:

Page 18 line 1:

"The SORT was the best-calibrated of the pre-existing models, however all over-predicted risk (Figure 2A-2C; Hosmer-Lemeshow p-values all "

HL test is not an appropriate approach to evaluate calibration in large cohorts where the frequency of the outcomes is low (See Tripod statements). Further a HL stat probability lower than 0.10 suggests mis-calibration. In the figure 2, HL probabilities are all lower than 0.0001. This somewhat confirms the visual analysis of the curves suggesting that the calibration of these scores is terrible (results already widely describe elsewhere)> 

Page 18 line 6:

"All models exhibited good-to-excellent discrimination (Figure 2D; AUROC SORT=0.91 (95% confidence interval (CI): 0.90-0.93); P-POSSUM=0.90, (95% CI: 0.88-0.92); SRS=0.85 (95% CI: 0.83-0.88)."

High observed discriminations are the consequence of:

1/ poor calibration of the models and

2/primary outcome #1.2%.

Presented CI seems to have been produced assuming a binormal distribution (the usual approach), this is not likely to be appropriate in this setting and the reader can guess that this CI are widely underestimated.

More importantly, no discrimination should be interpreted with such a poor calibration. 

Figure 5:

This figure looks great, but some confidence interval would probably show that there are some major overlaps between the c.

clinical prediction strata

Thank you again for your work and I hope my comments would be useful in your work.

There is a true need for preoperative stratification/predictive tools and this work has the potential to fill a part of this need

Yannick Le Manach MD PhD

Reviewer #4: The manuscript entitled „Comparing subjective and objective risk assessment for predicting mortality after major surgery - an international prospective cohort study" was reviewed. In this study, the authors aimed to compare subjective and objective risk assessment tools, which are utilized in predicting the probability of the postoperative mortality. The study was a well-designed, prospective, multicentric, observational study, with great effort to reduce the bias as much as possible, in terms of recruitment phase and data analysis. However, I there are some concerns regarding the manuscript, which should be addressed.

Comments:

1. In a recent study, Chan et al. (1) calculated the derivation and validation cohorts for mortality using the Combined Assessment of Risk Encountered in Surgery (CARES) surgical risk calculator, which is also comparable to the present study outcomes. The authors should also discuss the findings of this study.

2. How did the authors define the needed cut-off values for classification of preoperative estimation of mortality risk?

3. I would suggest the authors to perform a subgroup and validation analyses by classifying the patients based on the type of surgery.

4. How would it be possible to combine the two objective and subjective assessments in clinical practice. The combination of the tolls has improved the outcomes for sure, but authors did not suggest any combined approach to establish an estimation using both risk assessments simultaneously.

Reference:

1. Chan DXH, Sim YE, Chan YH, Poopalalingam R, Abdullah HR. Development of the Combined Assessment of Risk Encountered in Surgery (CARES) surgical risk calculator for prediction of postsurgical mortality and need for intensive care unit admission risk: a single-center retrospective study. BMJ open. 2018;8(3):e019427.

[LINK]

---

## [Decision Letter · Decision Letter 1]

18 Jun 2020

Dear Dr. Moonesinghe,

Thank you very much for re-submitting your manuscript "Developing and validating subjective and objective risk assessment measures for predicting mortality after major surgery: an international prospective cohort study" (PMEDICINE-D-19-04548R1) for review by PLOS Medicine.

I have discussed the paper with my colleagues and the academic editor and it was also seen again by previous reviewers. I am pleased to say that provided the remaining editorial and production issues are dealt with we are planning to accept the paper for publication in the journal.

[LINK]

We look forward to receiving the revised manuscript by Jun 25 2020 11:59PM. 

Sincerely,

Clare Stone, PhD

Managing Editor 

PLOS Medicine

plosmedicine.org

Requests from Editors:

Please provide summary demographic information to the abstract as previously requested. 

We suggest quoting AUROC/95% CI in the abstract for all the objective models. 

Please add a new final sentence to the "methods and findings" subsection of your abstract to quote 2-3 of the study's main limitations. 

Please remove the instructions from the "author summary".

Please convert p<0.0001 to p<0.001 throughout.

Please move the reference call-outs to precede punctuation (e.g., "... decision making [2,3].").

Is reference 2 missing full access details? 

Comments from Reviewers:

Reviewer #1: We thank the authors for addressing most of the points raised previously, and particularly appreciate the addition of a fourth sensitivity analysis on P-POSSUM, and an informative confusion matrix at 5% mortality. On the confusion matrix (Supplementary Table S6), we agree with the authors that a single cut-off ROC value is inadequate to characterize the ROC profile; the intention was chiefly to gain some additional perspective on the performance of the tool, which does appear to fulfil expectations. The authors might however note that some entries in the matrix appear slightly off-by-one (e.g. the Dead column is stated to total 189, but the sum of the six categories appears to be 188; 2.6-5% is stated to total 891, but the Alive+Dead for that row appears to be 892)

On part of the second point raised in the previous review on different methods having different numbers of point estimates in Figure 2 and 3, the authors' clarification on clinicians having 4 points in their calibration graph was much appreciated. However, given that it is stated that the predictions for P-POSSUM, SRS and SORT are continuous variables, the authors may wish to briefly comment on the different number of sampled points for these methods, as shown in Figure 2.

On the third point raised in the previous review ("additional inter-model analysis at finer granularity"), the major concern was whether the various risk models tend to have similar risk predictions for the same patients (in aggregate), and if not, what were the characteristics of patients that tended to be assigned different risks by different models. We agree with the authors that this may not be critical to the main thrust of the manuscript; it was proposed largely as the data appears available, and since it might be of interest given that multiple models are already being compared. As such, we would respect the authors' preference on whether to include such an analysis.

Minor issue: On page 7 line 10, "and equipoise over method is most accurate" might be "...which method is most accurate"

Reviewer #3: Dear Authors

Thank you for the responses to my comments. Regarding the limitations associated with the datasets and the objectives of your study, I believe you significantly improved your manuscript. 

I still disagree with the value of subjective score in some countries where these scores are used for billing purpose. It seems it was not the case in your cohort. However, I would be prudent in any generalization tentative. A. Sankar et al. (BJA Volume 113, Issue 3, September 2014, Pages 424-432) provided a report of this problem in a jurisdiction where ASA is used for billing.

While I will not ask for any revision for this version of the manuscript. I would be delighted if the authors would consider to add a sentence about the need for further researches since objective quantifications of the risk did not perform as well as when a subjective component is included (i.e. current models and approaches do not capture this part of the information provided by clinicians…more research needed to capture it in a more objective manner).

Thanks for considering

Yannick Le Manach MD PhD

[LINK]

---

## [Editor Report · Decision Letter 2]

3 Sep 2020

Dear Prof. Moonesinghe, 

On behalf of my colleagues and the academic editor, Dr. David Menon, I am delighted to inform you that your manuscript entitled "Developing and validating subjective and objective risk assessment measures for predicting mortality after major surgery: an international prospective cohort study" (PMEDICINE-D-19-04548R2) has been accepted for publication in PLOS Medicine. 

PRODUCTION PROCESS

PRESS

PROFILE INFORMATION

Thank you again for submitting the manuscript to PLOS Medicine. We look forward to publishing it. 

Best wishes, 

Clare Stone, PhD

Managing Editor 

PLOS Medicine

plosmedicine.org